# Stochastic Competition Networks for Deep Learning on Tabular Data

## Abstract

Despite the prevalence and significance of tabular data across numerous industries and fields, it has been relatively underexplored in the realm of deep learning. Even today, neural networks are often overshadowed by techniques such as gradient-boosted decision trees (GBDT). However, recent models are beginning to close this gap, outperforming GBDT in various setups and garnering increased attention in the field. Drawing from this inspiration, in this work we introduce a novel deep learning model specifically designed for tabular data. The foundation of this model is a Transformer-based architecture, carefully adapted to cater to the unique properties of tabular data through strategic architectural modifications, mainly two forms of stochastic competition. First, we employ the "Local Winner Takes All" mechanism as a refined alternative to ReLU-activated layers. Second, we introduce a novel embedding layer that blends multiple linear embedding layers through a form of stochastic competition. Model effectiveness is validated on a variety of widely-used, publicly available datasets. We show that, through incorporation of the said stochastic elements, we yield state-of-the-art performance and mark a significant advancement in applying deep learning to tabular data.

## 1 Introduction

Tabular data is a fundamental and arguably one of the most commonly used formats in the fields of data science and machine learning. It is structured with rows and columns that represent individual observations and their corresponding features; this creates a simple two-dimensional, table-like body. Within it, various data types can be included. This format enjoys widespread popularity in sectors like healthcare, finance, and sciences because of its organizational clarity and its close ties with relational databases and spreadsheets. Yet, despite its prevalence and seeming simplicity, effectively modeling tabular data for common tasks like regression or classification continues to pose significant challenges.

Features in tabular data can take several forms, ranging from simple scalar values to custom data structures. However, in modeling scenarios, these features predominantly manifest as either continuous real values or discrete categorical variables, often encoded as positive integers. Formally, a tabular row of length $s$ can be represented as $x \in \mathbb{R}^{s_r} \times \mathbb{N}^{s_n}$, where $s = s_r + s_n$. Here, $s_r$ demarcates the number of continuous features, while $s_n$ enumerates the categorical ones. Additionally, the positioning of features in a tabular row holds no intrinsic geometrical meaning. Thus, we presume no inherent relations between features, in contrast to other popular data forms like images or language.

Typically, the methodologies favored for these tasks have been Tree-Based models like Random Forests (Ho, 1995) and Gradient Boosted Decision Trees (GBDTs) (Friedman, 2002), chosen for their performance. Deep Learning, a paradigm that has substantially revolutionized learning for other forms of data, has not yet been established as the first line approach for tabular data. However, this trend is gradually evolving. Recent years have witnessed the emergence of few novel deep learning models that outperform GBDTs on a range of tabular datasets. While the related publications are still limited in number, their promising results have the potential to redefine the current approach to tabular data; this may position deep learning at the forefront of tabular data analysis.

In this paper, we delve further into the field, and propose a novel Deep Learning architecture for addressing Tabular Data. In particular, we consider the Transformer encoder (Vaswani et al., 2017)

as our architectural basis. We proceed to substantial modifications of this architecture to address tabular data, by adding: (i) a parallel fully connected element and (ii) an attention bias term. In addition, we infuse sophisticated stochastic techniques into the model, namely the powerful stochastic "Local Winner Takes All" (LWTA) layer, as well as a novel Mixture Embedding layer for the input features.

The remainder of this paper is organized as follows. The next Section offers an overview of related work. In Section 3, we introduce the proposed approach, explain its main architectural assumptions and components, and derive the training and inference algorithms. Section 4 provides a deep experimental evaluation of our proposed approach, using established benchmarks in the field; this is combined with a long ablation study. Finally, in Section 5 we conclude this paper drawing some key insights.

## 2 RELATED WORK

As previously outlined, the most established methods in Tabular Data Modeling (TDM) currently belong to the family of tree-based algorithms, especially in the form of Gradient Boosted Decision Trees (GBDT). These algorithms rely on an ensemble of weak learners, sequentially generated as corrections to the existing ensembles in a gradient-driven fashion. The most renowned and popular variants of such algorithms include Catboost(Prokhorenkova et al., 2018), XGBoost(Chen et al., 2015), NGboostDuan et al. (2020), and LightBoost(Ke et al., 2017). The popularity of these approaches, especially in industrial and competition environments, stems from their high performance and ease of use.

Until the close of the previous decade, deep learning methodologies for Tabular Data predominantly centered around multi-layer perceptrons and similar rudimentary architectures. However, the recent years have witnessed a surge in sophisticated neural network designs, yielding remarkable results. These contemporary designs have adopted diverse strategies, including emulating decision trees or other types of weak learners; often, they draw inspiration from GBDT. Two seminal architectures embodying this philosophy are NODE(Popov et al., 2019) and GrowNet(Badirli et al., 2020).

While these methodologies have recorded commendable outcomes, the trajectory in recent research has been the inclination towards Transformer-based architectures. Designs like TabNet(Arik & Pfister, 2021) harness the computational prowess of the Transformer and the attention mechanism, giving strong results through an encoder-decoder framework. Conversely, TabTransformer(Huang et al., 2020) deploys the transformer to process categorical tabular features and subsequently amalgamates the resultant representations with a fully connected layer to address the numerical features. FtTransformer(Gorishniy et al., 2021), meanwhile, employs an encoder-only design to analyze all features, and post-projects individual categorical features into distinct vector representations using a simple yet effective linear embedding layer. Finally, SAINT (Somepalli et al., 2021) goes beyond row-by-row processing through the addition of an inter-row attention layer.

Apart from proposing sophisticated network architectures, a number of studies have investigated the implications tied to distinct attributes and settings that underpin deep learning practices. Characteristic studies in this context consider pretraining (Rubachev et al., 2022; Iida et al., 2021), as well as various embedding approaches that yield strong results even with simple architectures such as MLP-PLR (Gorishniy et al., 2022). Standout contributions include Kotelnikov et al. (2023), which employs denoising diffusion probabilistic models, and Gorishniy et al. (2023), which employs retrieval augmentation strategies.

## 3 THE MODEL

### 3.1 OVERVIEW

In Figure 1, we provide a comprehensive overview of the proposed model, which employs a hybrid architecture grounded on an encoder-only Transformer. This foundational architecture is augmented with stochastic elements and additional structural modifications, which we will discuss in greater detail later in this Section.

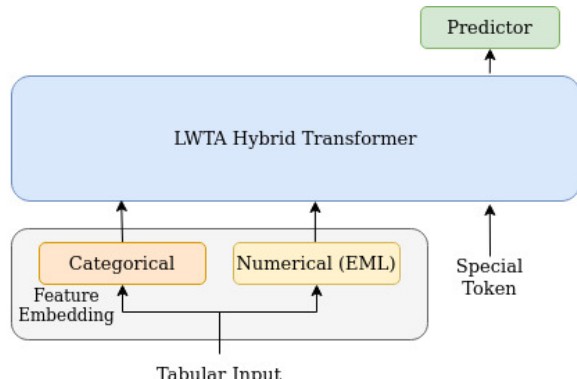

Figure 1: Overview of the proposed approach, exhibiting its core modules.

Our proposed adaptations do not obliterate the necessity for a specific input structure compatible with the standard Transformer encoder. To achieve compatibility with this structure, our first step is to adapt the original data format, defined in $\mathbb{R}^{s_r} \times \mathbb{N}^{s_n}$, to one that fits the Transformer. Through embedding layers, each feature $x_i, i \in 1, .., s$, be it numerical or categorical, is mapped onto a d-dimensional representation vector, given by $\boldsymbol{h}_i \in \mathbb{R}^d$. Eventually, a given input $\boldsymbol{x} = (x)_{i=1}^s$ is mapped to a vector $\boldsymbol{h} \in \mathbb{R}^{d \cdot s}$. Alongside this representation, we also add a vector, $\boldsymbol{h}_{special} \in \mathbb{R}^d$, that corresponds to an artificial "special token" with a static input value. The terminal representation of this token is fed to a penultimate regression or classification head, depending on the modeling task at hand.

While our architectural design shares similarities with usual Transformers and preceding models from TDM and other domains, it distinguishes itself through three key innovations that enhance its predictive capability: i) The adoption of the sophisticated stochastic LWTA layer (Panousis et al., 2019). The latter has been shown to yield improved results in a wide range of applications; yet, it has never been employed to networks designed for Tabular Data. ii) The introduction of a novel data-driven probabilistic selection among alternative (linear) feature embeddings. This enhancement adds an extra element of stochasticity and promotes richer feature representations. iii) The introduction of the Hybrid Transformer module, which is specifically designed for TDM. This module merges the core Transformer encoder with a parallel fully connected aggregation module. Tailored to capitalize on the static structure of tabular data, this aggregation module works by projecting the hidden representations back to scalar values and processing the aggregate result.

In the following subsections, we elaborate on each of the core novel elements that compose our Transformer-based approach.

### 3.2 LOCAL WINNER TAKES ALL

The Stochastic "Local Winner Takes All" (LWTA) layer (Panousis et al., 2019) is a more sophisticated alternative to common deterministic non-linear layers. This approach has garnered significant success in a range of tasks and setups, including Image Classification (Panousis et al., 2022), Meta-Learning (Kalais & Chatzis, 2022), Sign Language Translation (Voskou et al., 2021; Gueuwou et al., 2023), and more (Panousis et al., 2021b;a). An LWTA layer comprises linear units and introduces non-linear behavior by means of stochastic competition within blocks of layer neurons. Within a block of competitors, only one neuron, labeled as the "winner," is activated; winner selection is probabilistic. All other neurons remain inactive and pass zero values.

In a more formal notation, let us consider the input and output vectors of a typical linear layer, denoted by $\boldsymbol{x} \in \mathbb{R}^J$ and $\boldsymbol{y} \in \mathbb{R}^H$ respectively, with the associated weight matrix denoted as $W \in \mathbb{R}^{J \times H}$. In the LWTA approach, the elements of $\boldsymbol{y}$ are partitioned into $K$ distinct, non-overlapping blocks, each containing $U$ elements. Concurrently, the weight matrix $W$ is restructured into $K$ separate submatrices. This gives us $\boldsymbol{y}_k \in \mathbb{R}^U$ and $W_k \in \mathbb{R}^{J \times U}$ for each block $k \in \{1, 2, \ldots, K\}$. Within each block, the output values compete against one another and only

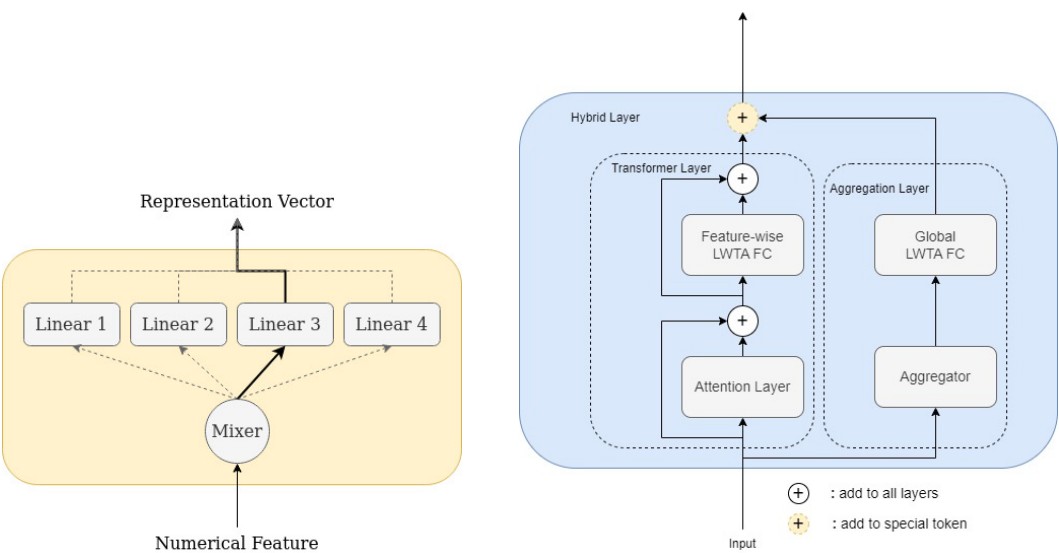

Figure 2: Illustration of the novel modules used in the proposed model: (Left) The embedding mixture layer. (Right) The hybrid Transformer module.

one, the "winner", is retained; the remaining elements are set to 0. The aforementioned competition is technically implemented as a stochastic sampling process inside each block. In this process, the winner indicator, a latent one-hot vector, $\boldsymbol{\xi}_k \in \text{onehot}(U)$ is sampled from a discrete posterior $D(\boldsymbol{\xi})$. The posterior logits are directly proportional to the linear computations of each respective unit, passed through a softmax. The final layer output $\boldsymbol{y}_k$ for the block $k$ is gained by using the postulated $\boldsymbol{\xi}_k$ in a simple masking operation as in (1).

$$\boldsymbol{y}_k = \boldsymbol{\xi}_k \odot (W_k \boldsymbol{x}), \ \xi_k \sim \text{D}\left(\boldsymbol{\xi}_k \Big| W_k \boldsymbol{x}\right), \ \ \forall k \in \{1, 2, ..K\} \tag{1}$$

where $\odot$ stands for element-wise multiplication. During training, $\boldsymbol{\xi}_k$ is approximated via a Gumbel-Softmax differentiable sample (Jang et al., 2016), to ensure effectiveness and stability:

$$\boldsymbol{\xi}_k = \frac{\exp\left((\log \eta_k + g_k)/T\right)}{\sum_{i=1}^{U} exp((\log \eta_i + g_i)/T)}$$
$$\boldsymbol{g} = -\log(-\log \boldsymbol{z}), \ \boldsymbol{z} \sim \text{U}(\boldsymbol{0}, \boldsymbol{1}) \tag{2}$$

where $\boldsymbol{\eta} = W_k \boldsymbol{x}$, and $T$ is a positive temperature hyperparameter.

### 3.3 Feature Embedding - Embedding Mixture Layer

Feature embedding serves as a pivotal element in models like the one we propose, acting as the bedrock upon which later processing stages are built. In our approach, each categorical feature is separately processed via a standard linear embedding layer. This technique is stable and well-grounded in the literature, sharing conceptual similarities with word embeddings commonly used in NLP.

Embedding of continuous values is much underexplored. Earlier work (Gorishniy et al., 2021) has mostly been limited to simple linear projections, computed independently for each feature. Recently, non-linear approaches have been explored and proved to be beneficial to the predictive accuracy (Gorishniy et al., 2022). In this work, we progress one step further, proposing a novel stochastic embedding layer that improves the expressive power of the vanilla approach. In our proposed method, instead of having a single pair of weight and bias vectors, we use a set of $J$ such pairs, defining $J$ alternative (linear) embeddings, each indicated by an indicator $j$. To gain the representation vector of a continuous input, $x_i$, the model has now to select one of the so-defined alternative linear projections. It does so in a stochastic manner, where the probability of one alternative embedding being

selected is driven by the value of $x_i$ via (3); this selection rationale is illustrated in Figure 2 (left side). We have

$$\boldsymbol{f}_{emb}(x_i) = x_i \cdot \boldsymbol{w}_j + \boldsymbol{b}_j, \quad j \sim \boldsymbol{P}(\cdot|x_i, \boldsymbol{\theta}_w, \boldsymbol{\theta}_b) \tag{3}$$

where the posterior probability distribution over the linear mapping reads

$$\boldsymbol{P}(j|x, \boldsymbol{\theta}_w, \boldsymbol{\theta}_b) = \frac{e^{t_j}}{\sum_{j=1}^{J} e^{t_j}}, \quad \boldsymbol{t} = x \cdot \boldsymbol{\theta}_w + \boldsymbol{\theta}_b \tag{4}$$

with $\boldsymbol{\theta}_w, \boldsymbol{\theta}_b \in \mathbb{R}^J$ denoting the trainable parameters directly involved in the selection process.

This embedding selection scheme can be described as a sort of competition among sub-parts at the embedding layer level; each competitor aims to dominate a broader range of input values. We posit that, in this way, the embedding engine can produce representations that are significantly richer than a single linear mapping. The eventually obtained embedding vector can vary considerably more than vanilla embeddings, based on the value regions of the input feature; this may allow for the identification of behavioral changes and shifts in statistical importance related to that feature. Additionally, the induced probabilistic transitions between different linear embeddings enhance accuracy in uncertain areas of mapping, and also help reduce the risk of overfitting.

While gating networks could be used to perform selection among alternative embeddings, our proposed method relies on competition, similar to stochastic LWTA layers. This is an effective alternative that obviates the need to introduce more trainable parameters for the gating function, and the associated computational burden. Similarly, we again utilize Gumbel-Softmax to provide a smooth, low-variance gradient during training.

### 3.4 Hybrid Transformer module

Typical Transformer input modalities, like text and videos, frequently display dimensionality that is subject to change, such as sentence lengths or video duration. Conversely, tabular datasets exhibit fixed, predefined dimensions. This distinct property offers an avenue for integrating static elements into the network, which would be unattainable in dynamically changing contexts. Our so-obtained hybrid Transformer module melds two essential sub-components. The first is a conventional Transformer encoder, which is a sequential arrangement of a Self-Attention layer and a Fully-Connected layer. In our work, we augment attention dot-product with a bias term, which we have empirically found to be a nuanced but effective adjustment. The second sub-component, which constitutes the novel aspect of our design, is a *parallel module*. This module can be technically described as an LWTA-based global Fully Connected Layer, as illustrated in Figure 2 (Right part).

The module is presented with the $d$-dimensional embeddings of each of the $s$ input features, reprojects them onto scalar values and aggregates them into a single $s$-dimensional representation vector through the operation $\Phi : \mathbb{R}^{d \cdot s} \rightarrow \mathbb{R}^s$, with

$$\Phi(\mathbf{h}) = (\boldsymbol{w}_i \cdot \boldsymbol{h}_i + b_i)_{i=1}^s \text{ where } \boldsymbol{w}_i, \boldsymbol{h}_i \in \mathbb{R}^d, b_i \in \mathbb{R} \tag{5}$$

The obtained consolidated vector, $\Phi(\mathbf{h})$, is presented to a subsequent LWTA layer, followed by a Linear layer; this yields an output vector $\boldsymbol{z} \in \mathbb{R}^d$. The output from this module is incorporated into the representation of the special token, in an additive (residual) manner.

### 3.5 Training and Inference

The training objective of our proposed model is formulated as follows:

$$\mathcal{L}(\phi) = \mathbb{E}_{q(\cdot)}\big[\log p(\mathcal{D}|\{\phi\})\big] - \text{KL}\big[Q(\{\boldsymbol{\xi}\}) \,||\, P(\{\boldsymbol{\xi}\})\big] - \text{KL}\big[Q(\{j\}) \,||\, P(\{j\})\big] \tag{6}$$

where $\{\boldsymbol{\xi}\}$ the set of the LWTA winner indicators, $\{j\}$ the embedding selection indicators and $\{\phi\}$ represents all the trainable parameters. It is captured by a composite functional consisting of three terms. The first term corresponds to the primary model objective. It incorporates the standard crossentropy loss for classification tasks and the mean squared loss for regression scenario. In both cases, the latent indicator vectors $\boldsymbol{\xi}$ and $j$ are replaced by a differentiable (reparameterized) expression obtained through the Gumbel-Softmax trick. The second term encapsulates the Kullback-Leibler divergences between the posteriors and the priors of the LWTA winner indicators, leveraging

Table 1: Key statistics and properties of benchmarking datasets.

| PART | HI | AD | OT | HE | JA | YE | DI | HO |
|---|---|---|---|---|---|---|---|---|
| Total Entries | 98049 | 48842 | 61878 | 65196 | 83733 | 515345 | 53940 | 22784 |
| Total Features | 28 | 14 | 93 | 27 | 54 | 90 | 9 | 16 |
| Catg Features | 0 | 8 | 0 | 0 | 0 | 0 | 3 | 0 |
| Task | C | C | C | C | C | R | R | R |
| Classes | 2 | 2 | 9 | 100 | 4 | – | – | – |

a uniform discrete prior distribution $U$:

$$\mathrm{KL}[Q(\boldsymbol{\xi})||P(\boldsymbol{\xi})] = \sum_{\forall \boldsymbol{\xi}} \sum_{i=1}^{U} Q(\xi_i) \log\left(Q(\xi_i)/U_i\right) \tag{7}$$

The third term is similar to the second, but quantifies the KL divergence between the posterior of embedding selection and a uniform discrete prior.

For model evaluation and inference, predictions are gained via Bayesian averaging. By executing the model multiple times, we average the resultant outputs from the employed classification or regression head.

## 4 EXPERIMENTAL RESULTS

### 4.1 BENCHMARKING DATASETS

In the experimental section of this study, we employ eight publicly available tabular datasets, in the same form as previously utilized in analogous research, such as Gorishniy et al. (2021), and Gorishniy et al. (2023). We use exactly the same train-validation-test split to facilitate fair comparison. Specifically, our analysis involves two datasets for binary classification, namely Higgs Small(HI) and Adult(AD); three datasets designed for multi-class classification, namely Otto Group Products(OT) with nine classes, Helena(HE) with 100 classes, and Jannis(JA) with four classes; and three datasets tailored to regression tasks, namely Year Prediction(YE), Dimanond(DI), and House16H(HO). As reference metrics, we follow a common practice and use Mean Squared Error for Regression and Accuracy for Classification Tasks.

The bulk of the selected datasets are medium-sized, with row counts ranging from 20,000 to 100,000. However, to also examine how performance changes when using a significantly larger dataset, we also use Year Prediction, a particularly popular dataset encompassing around half a million features. In the context of feature types, the majority of datasets include numerical attributes, with feature dimensions ranging between 5 and 93. Exceptions to this pattern are the Adult and Diamond datasets, which additionally incorporate categorical features. Detailed analysis of data statistics is provided in Table 1.

### 4.2 EXPERIMENTAL SETUP

In all experiments, the AdamW (Loshchilov & Hutter, 2017) optimization algorithm was selected, with a small weight decay rate, $wd \leq 10^{-4}$. Training was divided into two sequential phases: a short initial warm-up featuring a small ascending learning rate, and a subsequent main training part. In the latter phase, the learning rate commenced at $lr = 10^{-3}$ and was subject to a 50% reduction upon reaching a performance plateau. Additional hyperparameters included a fixed LWTA block size $U = 2$, as suggested by the majority of related literature (Panousis et al., 2019; Kalais & Chatzis, 2022) and confirmed in preliminary analyses; an mc-dropout rate of $p = 0.1 - 0.25$; and a Gumbel Softmax temperature $T = 0.69$ for training and $T = 0.01$ for inference. As usual with Gumbel-Softmax reparameterization, it suffices that we consider sample size $N = 1$ for training; we draw $N = 64$ samples for inference. Multi-head attention was incorporated with 8 heads. For input data prepossessing, appropriate normalization/scaling was employed, except for the OT dataset where original scaling was retained as suggested in Gorishniy et al. (2021). Additionally, we re-scale

Table 2: Results comparison with related Deep Neural Networks.

| Model | Classification ( Acc ↑) | | | | | Regression ( MSE ↓) | | |
|---|---|---|---|---|---|---|---|---|
| | HI | AD | OT | HE | JA | YE | DI | HO |
| MLP | 71.9% | 85.3% | 81.6% | 38.3% | 71.9% | 78.37 | 1.96 | 9.6845 |
| MLP-PLR | 72.9% | **87.0%** | 81.9% | – | – | – | **1.796** | **9.339** |
| Node | 72.6% | 85.8% | – | 35.9% | 72.7% | 76.40 | – | – |
| FtTransformer | 73.0% | 85.9% | 81.7% | 39.1% | 73.2% | 78.40 | – | 10.48 |
| Saint | 72.9% | 86.0% | 81.2% | – | – | – | 1.877 | 10.51 |
| STab | **73.3%** | 86.1% | **82.5%** | **39.5%** | **73.5%** | **76.10** | 1.825 | 9.550 |

Table 3: Results comparison with Gradient Boosted Decision Trees and ensemble models.

| Model | Classification ( Acc ↑) | | | | | Regression ( MSE ↓) | | |
|---|---|---|---|---|---|---|---|---|
| | HI | AD | OT | HE | JA | YE | DI | HO |
| XGBoost | 72.6% | **87.2%** | 83.0% | 37.5% | 72.1% | 79.98 | 1.877 | 10.09 |
| XGBoost$_{ens}$ | 72.8% | **87.2%** | **83.2%** | 38.8% | 72.4% | 78.49 | 1.850 | 10.00 |
| CATBoost | 72.6% | 87.1% | 82.5% | 38.5% | 72.3% | 78.98 | 1.796 | 9.720 |
| CATBoost$_{ens}$ | 72.9% | **87.2%** | 82.7% | 37.7% | 72.7% | 78.11 | **1.769** | 9.645 |
| MLP-PLR$_{ens}$ | 73.5% | **87.2%** | 82.2% | – | – | – | 1.769 | **8.958** |
| Node$_{ens}$ | 72.7% | 86.0% | – | 36.1% | 73.0% | 76.02 | – | – |
| FtTransformer$_{ens}$ | 73.3% | 86.0% | 82.4% | 39.8% | 73.9% | 76.51 | – | 10.17 |
| STab$_{ens}$ | **73.6%** | 86.2% | **83.2%** | **40.0%** | **74.0%** | **75.60** | 1.781 | 9.300 |

the labels of HO and DI by a factor of $10^{-4}$ and $10^2$, respectively for better illustration purposes. All reported results regarding the proposed method correspond to the average of 4 different trainings from different random seeds; all ensemble scores are combinations of these 4 runs. All computations were executed on a single 24GB GPU.

## 4.3 RESULT DISCUSSION

Table 2 presents a comparative evaluation of our proposed model against leading deep-learning benchmarks, specifically MLP-PLR, NODE, FtTransformer, and SAINT, as well as a basic MLP. To maintain a focused examination of architectural differences, we intentionally exclude methods that rely on transfer learning or data augmentation. For the proposed model (STab), we employ our recommended hyperparameters, to be detailed later in this section, and perform inference through Bayesian averaging with a sample size of $N = 64$. For established benchmarks, we cite results from existing literature as provided in Gorishniy et al. (2021); Rubachev et al. (2022); Gorishniy et al. (2023), or Somepalli et al. (2021). This approach not only conserves computational resources but also ensures impartiality through third-party verification of performance metrics.

Our model demonstrates superior performance, outperforming existing neural network architectures in 5 out of the 8 evaluated benchmarks. Exceptions occur in the HO, AD and DI datasets, where our model still performs very well and ranks second, trailing only behind MLP-PLR. In a more comprehensive analysis, presented in Table 3, we extend the comparison to include ensemble models as well as two established GBDT paradigms in both single and ensemble configurations. While our model's superiority persists in ensemble settings, the margin of lead narrows slightly. Gradient Boosting models in their ensemble form closely align with our results on the OT task, and CATBoost's marginally outperform us on DI. In addition, our model seems to benefit slightly less from ensembling compared to some older deterministic deep networks, possibly due to its inference mechanism via Bayesian averaging. Nonetheless, the ensemble version of our model remains the state-of-the-art solution for the majority of the evaluated tasks.

In Table 4, we list the main hyper-parameters of the proposed model for each dataset, corresponding to the experimental results presented. These values might not showcase the absolute best performance, as we opted against exhaustive optimization. Additionally in situations with marginally differing results, factors such as model size were also taken into consideration.

Table 4: Suggested hyperparameters

| Model | HI | AD | OT | HE | JA | YE | DI | HO |
|---|---|---|---|---|---|---|---|---|
| Dropout | 0.225 | 0.1 | 0.25 | 0.25 | 0.25 | 0.25 | 0.1 | 0.125 |
| Embedding J | 4 | 16 | 16 | 16 | 64 | 16 | 16 | 16 |
| Embedding Size | 176 | 16 | 180 | 96 | 192 | 128 | 96 | 128 |
| Depth | 4 | 3 | 5 | 7 | 4 | 6 | 4 | 4 |

Table 5: The effect of mixture embedding parameter $J$ (upper) and LWTA block size $U$ (lower).

| | HI($\uparrow$) | HE($\uparrow$) | DI($\downarrow$) | HO($\downarrow$) |
|---|---|---|---|---|
| J= 64 | 73.2% | 39.4% | 1.84 | 9.94 |
| J= 16 | 73.2% | 39.5% | 1.83 | 9.55 |
| J= 4 | 73.3% | 39.5% | 1.84 | 9.67 |
| J= 1 | 73.2% | 39.1% | 1.87 | 9.88 |
| U= 4 | 73.2% | 39.2% | 1.83 | 9.51 |
| U= 2 | 73.3% | 39.5% | 1.83 | 9.55 |

## 4.4 ABLATION STUDY

### 4.4.1 EMBEDDING MIXTURE PARAMETER $J$

To evaluate the influence of the Probabilistic Embedding Mixture on our model's performance, we conducted a specific study, the results of which are displayed in the upper section of table 5. This analysis concentrates on the significance of parameter $J$. It is important to note that using $J = 1$, is equivalent to using a standard linear embedding. Data from tables 5 and 4 indicate that, in many instances, $J = 16$ appears to be the optimal value. Yet, in most cases, slight adjustments in $J$, whether below or above the optimal, don't lead to significant changes in performance metrics. Despite this, there is a noticeable improvement over the standard linear numerical feature embedding.

### 4.4.2 LWTA BLOCK SIZE $U$

To support our decision to maintain a constant LWTA block size of $U = 2$, not solely based on prior literature, we provide a brief analysis on the effects of a higher block size in the lower segment of table 5. The findings suggest that increasing the block size, such as to $U = 4$, usually results in either subpar performance or only a minimal effect. These findings further validate our selection of $U = 2$ as an effective default value, lessening the impetus for further investigation.

### 4.4.3 BAYESIAN AVERAGING AND SAMPLE SIZE

Due to the inherent stochastic nature of our model, we obtain its final prediction through Bayesian averaging. In Figure 3, we examine the relationship between sample size and prediction quality. As it was expected, we find that increasing the sample size generally improves and stabilizes the prediction, which yet starts reaching a plateau for around $N = 20$.

While our averaging approach may look similar to model ensembling, it's crucial to point out that they differ in key aspects. Unlike model ensembling, which requires training multiple (N) distinct models, our method needs just a single model to be trained. This means no need for extended training processes neither additional memory and storage space. Additionally, while it is true that inference time increases linearly with N in either case, this does not hold for single-row inference or small batches. In these cases, even for very large N, drawing N samples can be performed in parallel on a single GPU without additional delays. This is particularly advantageous for real-time applications requiring low latency and rapid response times.

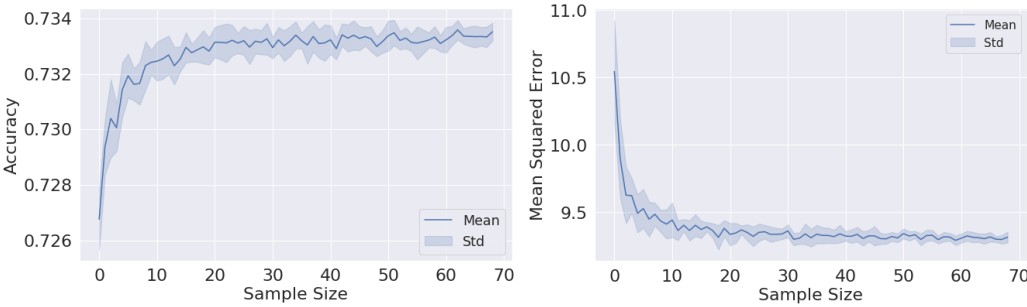

Figure 3: The effect of Sample size N on model's performance : (Left) Accuracy Higgs boson detection, (Rigth) Mean Squared Error on House16H

## 5 LIMITATIONS

While the proposed model achieved S.O.T.A. results in the majority of the tasks, it does not demonstrate a global domination, although no model has achieved this until now. Additionally, similar to most deep networks in current literature, it has a complex structure and requires much more resources for both training and inference than the popular GBDT.

## 6 CONCLUSION

In this study, we introduce a novel approach to tabular data modelling by harnessing contemporary deep learning, with a particular emphasis on stochastic competition techniques. We employ a stochastic Transformer-based model with a modified task-adapted architecture. The model's computational prowess is further augmented by the integration of the stochastic LWTA layer. Additionally, we unveil a distinctive embedding mixture layer for numerical features, seamlessly fusing multiple linear mappings through a stochastic competition mechanism. As a testament to our approach's efficacy, we secured state-of-the-art results on a majority of eight popular benchmarks and achieved second place among recent deep learning methodologies in the remaining instances. Notably, these advantages persist even in ensemble model configurations.

In upcoming research endeavors, we recommend a thorough exploration of stochastic competition methods, with the goal of enhancing model performance for tabular data and setting the stage for a deep learning framework in this GBDT-dominated area. Another avenue of interest is understanding how these stochastic techniques can leverage sample outcomes to estimate metrics beyond just expected values; this includes assessing uncertainties and probing into the distributional aspects of predictions. Also, incorporating advanced strategies, such as smart data augmentation, transfer learning, and meta-learning, offers a promising perspective for future studies. Historically, these methodologies have demonstrated their effectiveness by markedly improving model outcomes, suggesting their potential to elevate the efficacy of our proposed architecture.

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
