# OpenReview forum: "Stochastic Competition Networks for Deep Learning on Tabular Data"
_ICLR.cc/2024/Conference — ICLR 2024 Conference Withdrawn Submission_

### Official Review · Reviewer_fLPM · 2023-10-14

**Soundness:** 2 fair
**Presentation:** 2 fair
**Contribution:** 1 poor
**Rating:** 3
**Confidence:** 5

**Summary:**

**Scope:** machine learning problems on tabular data (e.g. classification, regression).

**Contribution:** *STab* -- a Transformer-like architecture for tabular data problems. There are four architectural elements that make STab different from a vanilla Transformer:
1. *LWTA* ("Local Winner Takes All") -- a stochastic activation used instead of the ReLU activation in feed-forward blocks.
2. *"Embedding mixture layer"* -- a LWTA-based embedding layer for scalar continuous features.
3. *A trainable bias* (shape `N_features x N_features`) is added to attention maps in all heads of all attention blocks.
4. *"Parallel module"* -- a LWTA-based feed-forward module in each Transformer block that that runs in parallel with the main block and enriches only the CLS embedding.

**The main claim:** *"we yield state-of-the-art performance and mark a significant advancement in applying deep learning to tabular data"*.

**Strengths:**

- The story is easy to follow.
- Developing better tabular DL architectures is an interesting research direction.
- Specifically, I appreciate that the paper aims at developing a variation of Transformer architecture that would be more tailored specifically for tabular data problems instead of being copied as-is from other fields, which can be a suboptimal approach.
- In particular, the bias term in the attention module is indeed such an element that is motivated by the nature of tabular data.
- I appreciate that the code is available, I managed to launch it

**Weaknesses:**

(1) Unfortunately, in my opinion, **the novelty is limited:**
- The LWTA "activation" is not a new element as such (in general, applying existing techniques is fine if well-motivated or supported by strong empirical performance).
- In particular, the "stochastic feature embeddings" is a `Linear-Activation-Linear` embeddings with LWTA used as `Activation`, which makes the proposed module very similar to the embeddings schemes analysed in "On Embeddings for Numerical Features in Tabular Deep Learning" by Gorishniy et al.
- Although I like the idea of adding a bias to the attention map, I should say that it was previously explored in other fields (e.g. see "Position Information in Transformers: An Overview" by Dufter et al.).
- The "parallel module" is technically new, but, subjectively, this is an incremental addition which is not enough to support the overall novelty.

(2) In my opinion, **the proposed architectural elements need more motivation** (*specific* observations/analysis/experiments/theory/citations; heuristic claims are fine, but, in my opinion, they are not enough). The world of deep learning offers many techniques that can be used to slightly improve the performance of existing tabular models. Then, the question arises: why specifically for tabular data problems should we use, for example, LWTA/"parallel module"/etc.? Without answering this question, it can be hard to break the inertia of using simple well-established elements.

(3) Subjectively, **the reported results are not fully in line with the main claim, and they do not compensate for other issues ((1), (2), and (4))**. Specifically, I refer to Table 2, Table 3, the reported margins, the number of wins, the benchmark size and the claim *"we yield state-of-the-art performance and mark a significant advancement in applying deep learning to tabular data"*.

(4) **The paper adds new non-trivial complexity** to a well-established Transformer architecture, which is not a problem as such, but, in my opinion, the previous issues should be resolved to motivate the community to use a new non-trivial architecture.

(5) Other smaller things in no particular order:
- In my opinion, some parts of the story could be more compact or could be moved to the appendix. For example, Table 4. Similarly, Section 4.1 could be more compact. Perhaps, in the introduction, the nature of tabular data could be also explained in a more compact way.
- I recommend increasing the resolution of the illustrations.
- I recommend supporting the story in the introduction with more citations.
- (Code) `STab/mainmodel.py`: there is a (currently, non-critical) bug on line 72: it should be `LWTA(U=U)`. I recommend tweaking the code editor to make it highlight such typing-related issues.
- (Code) I tried launching `TrainAD.ipynb` but obtained suboptimal results. I appreciate that the code is marked as "experimental", but I am reporting this just in case. It seems that the number of epochs is too low.
- I recommend proof-reading the paper for English style, vocabulary and grammar issues.
- I see the place where the "TDM" abbreviation was introduced, but overall, this is the first time I see this abbreviation (this can be my fault though). Perhaps, it can be avoided.

**Questions:**

How negative values in $\eta$ are handled in Equation 2?

---

### Official Review · Reviewer_Nj1J · 2023-10-26

**Soundness:** 2 fair
**Presentation:** 3 good
**Contribution:** 2 fair
**Rating:** 3
**Confidence:** 4

**Summary:**

The authors propose an interesting transformer architecture, that features 3 extra components. The first is a change for the linear layer in the transformer layer/block that is subtituted with an LWTA layer, which I would phrase as a "guided dropout". In the same way, a similar procedure is applied before feeding the data to the transformer, where LWTA is applied but in this case with layers and not distinct blocks inside a layer. Lastly, a parallel module is added to the transformer layer/block which projects the feature embeddings and then aggregates them, adding them to the output of the transformer layer.

The authors compare the proposed method against different deep learning baselines and tree-based methods on 8 diverse datasets (proposed from prior work) covering binary/multiclass classification and regression.

**Strengths:**

- The paper is written well and it has a good structure.
- An ablation is performed for one of the proposed components.

**Weaknesses:**

- **Until the close of the previous decade, deep learning methodologies for Tabular Data predominantly
centered around multi-layer perceptrons and similar rudimentary architectures. However, the recent
years have witnessed a surge in sophisticated neural network designs, yielding remarkable results.**

    I would argue against the above paragraph, Kadra et al. (2021) [1] have shown that tabular resnets
    when carefully regularized manage to outperform specialized deep-learning architectures and tree-based methods.
    An outcome that is further verified by Shwartz-Ziv et al. (2022) and Gorishniy et al. (2021) [2][3].

    In this regard, the related work is not extensive and can be further improved.
- I could not find an ablation of the LWTA instead of the linear layer as a component for the transformer layer. There additionally is no ablation regarding the proposed parallel module that projects the feature embeddings and aggregates them, finally summing them with the transformer layer.
- The authors claim state-of-the-art results with only 8 datasets.
- The authors reuse results for the baselines, which can potentially propagate failures in the setup of competitor methods, depending on where the results were taken. At the same time, it is not clearly indicated which results were taken from where.
- It is not clear whether the authors use default hyperparameters for the baselines or tuned hyperparameters by reading the core manuscript. For the proposed methods the hyperparameters are tuned, however no information is given on what procedure was used and how much time it took.
- No information is provided regarding the runtime of the proposed method or the competitor baselines.

[1] Kadra, Arlind, et al. "Well-tuned simple nets excel on tabular datasets." Advances in neural information processing systems 34 (2021): 23928-23941.

[2] Shwartz-Ziv, Ravid, and Amitai Armon. "Tabular data: Deep learning is not all you need." Information Fusion 81 (2022): 84-90.

[3] Gorishniy, Yury, et al. "Revisiting deep learning models for tabular data." Advances in Neural Information Processing Systems 34 (2021): 18932-18943.

**Questions:**

- Should the numerator in Equation 2 not go through a specific $u$ value, it is slightly confusing as $k$ is used for the distinct blocks, however, $\xi_k$ represents the one hot vector inside a block of $u$ units if I am not mistaken.

- **For established benchmarks, we cite results from
existing literature as provided in Gorishniy et al. (2021); Rubachev et al. (2022); Gorishniy et al.
(2023), or Somepalli et al. (2021). This approach not only conserves computational resources but
also ensures impartiality through third-party verification of performance metrics.**

    Could the authors describe for which baselines were the results reused? I think it would be beneficial that all methods be run by the authors since in many cases, it can happen that competitor baselines can be run in the wrong way.

- **In Table 4, we list the main hyper-parameters of the proposed model for each dataset, corresponding to the experimental results presented.**

    Pending on my previous question, I assume the authors at least partially (if not for all) have not performed hyperparameter tuning for the considered baselines (since they reuse results). In contrast, they have performed hyperparameter tuning for their method.

- Could the authors provide an ablation of all the proposed components? It would be nice if it could be provided for all datasets and as distribution plots for the different values over the datasets, that way it would be easier to observe the differences.
- Could the authors provide the ranks that all methods achieve over the datasets to aggregate the results?
- It would be interesting to compare the runtimes of the different methods.

---

### Official Review · Reviewer_Fv9b · 2023-10-29

**Soundness:** 2 fair
**Presentation:** 1 poor
**Contribution:** 2 fair
**Rating:** 3
**Confidence:** 2

**Summary:**

This research addresses the underexplored use of deep learning in handling tabular data, a prevalent and significant data format in various industries. Historically, techniques like gradient-boosted decision trees (GBDT) have dominated this field. However, recent models are narrowing this gap, surpassing GBDT in different scenarios and gaining increased attention. The study introduces a novel deep learning model designed specifically for tabular data, using a Transformer-based architecture that has been modified to accommodate the unique characteristics of tabular data. These modifications include incorporating elements like "Local Winner Takes All" mechanisms and a novel Mixture Embedding layer.

**Strengths:**

It seems to me that the Mixture Embedding layer is new.

**Weaknesses:**

Four runs were conducted, but standard deviation data is not available. The AUC is also considered a relevant metric. Overall, I don't see the justification for using more complex models with significant resource requirements over a less interpretable model like XGBoost. The figures have low quality; please consider using 'TikZ' or 'tikzpicture' in LaTeX for better quality figures.

**Questions:**

see above

**Details Of Ethics Concerns:**

-

---

### Official Review · Reviewer_tizg · 2023-10-31

**Soundness:** 2 fair
**Presentation:** 2 fair
**Contribution:** 2 fair
**Rating:** 5
**Confidence:** 3

**Summary:**

In this work, the authors propose an approach to apply a vision transformer architecture to tabular data. More specifically, the authors introduce in a transformer-based architecture three mechanisms. The first is the "Local winner takes all" previously proposed by Panousis et al., which applies a heavy non-linearity to select the firing output in parallel branches and smoothed through the Gumbel softmax. Then, an embedding mixture layer is introduced, and finally, the hybrid transformer module is assembled. The experimental benchmarks are inspired by similar works in the literature.

**Strengths:**

- Deep learning-based techniques for tabular data are not very popular, and this work goes in this direction
- the paper is clear in most of its parts

**Weaknesses:**

- the ablation study is incomplete
- the confidence intervals in the main results are missing- this questions the superiority of the proposed approach
- the model and the training complexity are in general not evaluated when results are presented
- some parts are not very clear and require some interpretation to be understood (Sec. 3.3)

**Questions:**

- can you provide the ablation where you show that every introduced component is clearly improving the performance of a vanilla transformer-based approach? The current ablation study does it just partially
- can you provide standard deviations for the main results?
- can you evaluate the trade-off between training/inference complexity and final performance? it feels that the proposed approach has longer training/inference time and occupies much more space, for a marginal improvement